# Analysis of Small Sea-Surface Targets Detection Performance According to Airborne Radar Parameters in Abnormal Weather Environments

**DOI:** 10.3390/s22093263

**Published:** 2022-04-24

**Authors:** Hamza Bounaceur, Ali Khenchaf, Jean-Marc Le Caillec

**Affiliations:** 1Lab-STICC UMR CNRS 6285, ENSTA Bretagne, 2 Rue François Verny, 29806 Brest, France; hamza.bounaceur@ensta-bretagne.org; 2Lab-STICC UMR CNRS 6285, IMT Atlantique, 655 Avenue du Technopôle, 29280 Plouzané, France; jm.lecaillec@telecom-bretagne.eu

**Keywords:** maritime scanning radar, UAV, signal-to-clutter ratio, sea clutter, target RCS, Doppler frequency, atmospheric attenuation

## Abstract

Along with the rapid development of marine radar, and particularly those carried on aircraft, the detection of small-sized targets which pose an increasing threat has become one of the main areas of interest. However, by considering an observation chain from an aircraft (such as a drone) in a maritime environment, with the aim of detecting and tracking of low signal-to-clutter ratio (SCR) targets, one of the important points would be the analysis of the radar system performance according to the radar input parameters, the atmospheric propagation medium, the various sea clutter characterization, and the type of targets (RCS, speed, etc.) in this environment. Therefore, it is necessary to obtain the overall path loss including the anomalous atmospheric environment, gas attenuation, clouds attenuation, rainfall attenuation, and beam scanning loss. To consider atmospheric attenuations, ITU-R models are used. On another side, because of spikes and dynamic variation properties, sea clutter is generally described by the statistical distribution with long tail and by its wider Doppler spectrum. Conventional algorithms such as those based on statistical models, MTI, and MTD processing are often limited, especially for the target of low speed and low RCS. Therefore, sea clutter, including empirical and statistical models available, is considered to estimate and simulate the impact of radar input parameters, targets RCS, and sea state on detection performance. The Doppler frequency of target echo which can be exploited for coherent processing is described by assuming an adequate scenario of observation geometry.

## 1. Introduction

With the world’s rapid economic and social development in recent years, great changes have been constantly taking place in the sea domain. Therefore, maritime area observation and marine search and rescue are becoming an important research issue and attract huge interest [1,2]. Thus, there is a strong demand in the observation and remote sensing community for data on interesting regions and to update geospatial information flexibly and quickly [3,4,5]. Most importantly, observation of maritime surface targets from aerial platforms such as those in Figure 1a requires scanning, identification of areas of interest, and deployment of suitable, adequate platforms and sensors to thwart the threats posed by fast and low reflectivity marine boats, due to their use in illegal activities such as piracy, smuggling, human trafficking and drug trafficking. 

Radars are the most used sensors for radio navigation, surveillance (land, air, sea), fire control, observation of natural environments (atmosphere, sea, snow cover, soil, subsoil, etc.), and because of the presence of various phenomena such as the degradation of the electromagnetic signal by the atmosphere and/or its reflection by undesirable obstacles, the observation of the maritime environment from an aircraft requires the mastery of at least three aspects: specificities and characteristics of the sensors, propagation channel (atmospheric and maritime), and innovative processing methods.

In addition to their all-weather capacity (day/night), airborne scanning radars located on satellites or manned aircraft often have the capacity to monitor large areas. However, these two platforms have certain limitations: lack of flexibility due to the orbit of satellites and area needed for planes to take off and land along the operational cost make their use difficult. Effective observation of the sea surface is starting to rely on Unmanned Aerial Vehicles (UAVs) [6,7,8]. Such aircrafts with limited payloads benefit greatly from the lightweight nature of low power millimeter wave radar systems, which offer good resolution from a small aperture. In contrast with manned aircraft and satellites, UAVs have many promising characteristics—flexibility, efficiency, high spatial/temporal resolution, low cost, easy operation, and so forth—which make them an effective complement to the other two platforms and a cost-effective means for remote sensing. Because of their unique advantages, types and number of UAVs are significantly increasing. People often classify them by size, range, speed, endurance, and altitude (Table 1). In addition, the development of UAV technology and advances in the small size, light weight, and high detection precision radars equipping these platforms make UAVs a popular and increasingly used technique in the field observation of the terrestrial environment [9,10,11] as well as the maritime environment [12].

In this paper, by considering an observation chain from a UAV, with the aim of detecting and tracking low RCS moving boats such as those in Figure 1b [14], one of the fundamental steps relates to the fine characterization of propagation environment or propagation channel in which the electromagnetic signal used by the radar is propagated and reflected. As shown in Figure 2, the propagation channel includes the atmosphere crossed by the radar signal in its round trip towards the targets of interest and the maritime environment surrounding these targets. Due to the components of the first propagation medium (atmosphere) and the physical nature of the second (sea surface), the propagation of the electromagnetic signal is subject to changes in intensity, frequency, and direction. Consequently, even if the returned signal from an object to be detected depends on the geometrical and physical natures of the object, it can often be degraded by atmospheric factors, thermal noise, and especially by sea surface backscattering. Furthermore, its frequency can be subject to a change applied by both the radar platform and object motion. It is therefore essential to know the mechanisms involved in the propagation as well as the radar waves interactions with the environment in order to be able to predict the detection performance and the tracking of small maritime targets, in particular in harsh weather conditions.

The observation chain at the level of the propagation environment should be analyzed assuming that the radar waves traveling to and from the targets are in free space and atmospheric gas. Even clear air is not lossless; its oxygen and water vapor both absorb energy. Although radar systems have the advantage of being able to “see through” cloud, fog, and rain, these weather factors degrade the strength of the radar signal. Hence, less energy reaches an object to be detected and less returns as target echo. Therefore, precipitation and hydrometeors as cloud and rain, when present, add an additional element which should be considered when assessing the likelihood of detecting an object. The usual effects of weather factors are to reduce the ranges at which targets can be detected and to produce false detections. The reduction of the radar signal intensity along its path is known as attenuation [15]. A detailed description of the radar signal attenuation by the atmosphere has been provided by Blake [16]. The amount of attenuation caused by each of the various factors depends, to a substantial degree, on the radar frequency.

To properly evaluate the performance of airborne maritime radar, it is critical to understand the propagation environment. The problem related to the attenuation of the signal in the atmosphere explains only part of the propagation, while the nature of the echoes backscattered by a sea surface is characterized by its own physical and geometric properties and is a greater impact factor in the propagation environment. In reality, detecting and tracking ships and small surface objects such as lifeboats, buoys, and castaways floating on the sea surface using a radar system has been proven to be a very challenging problem. Detection involves determining whether or not there is an object located in the sea surface of interest. Tracking is more sophisticated than detection because in addition to detecting an object, tracking systems attempt to calculate distance and/or radial speed of the target by processing the return signal in the presence of sea echoes. Usually, this kind of backscattering is referred to as sea clutter [17]. Its characteristics, specifically the mean backscattering, Doppler spectrum [18], and spikiness or amplitude statistics are known to vary according to the radar parameters, the viewing geometry, and weather conditions. They impact the detection ability to detect some targets. In fact, reflected echoes from small boats of interest, for which the Radar Cross-Section (RCS) is from −10 dBsm (0.1 m²) for small wrecked-vessel parts up to 6 dBsm (4 m²) for medium-size lifeboat or raft, is generally less than typical sea clutter echoes, which, in typical cases, varies from −30 dBsm for calm sea up to 10 dBsm for high sea state [19], and hence, they are undetectable. On the other hand, in addition to the radar platform motion, the Doppler spectrum is widened by Doppler spreading due to the perpetual movement of sea water. Therefore, the problem of detecting low RCS fixed or slowly moving objects, with a range velocity smaller than Doppler velocity resolution cell, is much more difficult, because these types of targets can be hidden within the sea Doppler spectrum. In general, discriminating between target echoes and unwanted echoes from the sea surface surrounding these targets is one of the major difficulties associated with airborne radar. Traditional methods such as traditional constant false alarm rate (CFAR) technologies [20,21,22] are often limited.

After the presentation of the problem of observing a maritime environment from an aircraft (of the UAV type), the problem dealt with in this paper relates to one of the elements of the chain, namely the characterization and the impact of the propagation channel on the performance of the radar positioned on the observation drone. The airborne radar is characterized by its observation geometry (movement of platform, antenna scan mode, and grazing angle) and its parameters (frequency, polarization, range resolution, azimuth resolution, etc.), which can be changed to improve radar performance according to the types of desired targets to be detected and the environment. The main goal in this paper is to evaluate the degradation of electromagnetic waves in different states of the atmosphere that this radar can potentially encounter, where the observation is carried out from a drone, and then to analyze the radar backscattered energy due to the complex interaction between these incident waves and the ocean surface to estimate both the amplitudes of clutter echo and the amplitude of target echo, because incoherent detections are mainly based on the analysis of the amplitude of the compound radar echo. We also estimate the variation of the Doppler frequency of the target according to an adequate observation geometry model, since in the case of a target with a weak signature, the coherent detection can preserve the phase information making it possible to differentiate the target from the sea clutter by its Doppler spectrum. In addition to this introduction, the rest of this paper is organized as follows: in Section 2, the amplitude factor models of both target echo and sea clutter echo are established. In Section 3, a critical analysis of the several models making it possible to estimate and simulate the degradation of electromagnetic signal in the atmosphere is performed. The discussion is based on the ITU-R P series recommendations of the International Telecommunication Union. The objective is also to analyze the performance of the overall link taking into account analytical and/or empirical models retained, dedicated to a better characterization of the sea scatter. Finally, depending on the appropriate observation geometry, i.e., the movement of the drone and the scanning mode of the radar it carries, a Doppler frequency study of a target is carried out according to the different scenarios of movement of this target in the scanning region. Section 5 concludes this article by presenting a summary of the results obtained and also the presentation of some future works carried out and presented in this paper.

## 2. Amplitude Factors of Target Echo and Sea Clutter

### 2.1. Echo Signal Model of Target

The mode of sending signal is assumed linear frequency modulation wave (LFMW), commonly known as linear chirp. It is the most commonly used waveform in radar systems as it can be easily generated by a variety of technologies. LFMW has a linear time frequency description as its frequency which varies linearly over the pulse duration of the signal and has high range resolution due to pulse compression. It can be expressed as follows.
(1)$$S_{t}\left(\eta \right)=u\left(\eta \right)\cdot e^{j2\pi \left(f\eta +\frac{B}{2\cdot T}\cdot \eta^{2}+\varphi_{t}\right)}\;\;0\leq \eta \leq T$$
where η is the fast time, f is the carrier frequency, *T* is the pulse duration, B  is the pulse bandwidth, $\varphi_{t}$ is the initial phase, and u(η) is the envelope of the radar signal.

The input of the radar receiver is the backscattering echo energy density reflected from a target. The echo power density $p_{r}=\left|S_{r}\left(\eta ,t\right)\right|²$ received by the receiving antenna is:(2)$$p_{r}=\frac{p_{t}G²\lambda ²}{{\left(4\pi \right)}^{3}R_{t}^{4}L_{r}L_{a}}\sigma_{t}$$

$p_{t}$: peak transmitted power (W), $\sigma_{t}$: radar cross section of target ($m^{2}$)G: single pass antenna power gain, *λ*: radar wavelength (m)$R_{t}$: instantaneous range to target (m), $L_{r}$: radar loss$L_{a}$: atmospheric and propagation loss. 

The echo signal of a moving target $S_{r}\left(\eta ,t\right)$ can be expressed as:(3)$$S_{r}\left(\eta ,t\right)=K\cdot u\left(\eta -\tau \left(t\right)\right)\cdot e^{j2\pi \left(f\left(\eta -\tau \left(t\right)\right)+\frac{B}{2\cdot T\;}\cdot \left(\eta -\tau \left(t\right)\right)²+\varphi_{r}\right)}$$
where t is the slow time and $\varphi_{r}$ is the echo phase.

K is the attenuation due to propagation:(4)$$K={\left[\frac{G²\lambda^{2}}{{\left(4\pi \right)}^{3}R_{t}^{4}L_{r}L_{a}}\cdot \sigma_{t}\right]}^{\frac{1}{2}}$$

τ(t) is the round-trip time delay of the signal from the radar to the target and back to the radar:(5)$$\tau \left(t\right)=\frac{2R_{t}\left(t\right)}{c}=\frac{2}{c}\left(R_{0}-V_{rad}\cdot t\right)\;$$

$R_{0}$ is initial range when *t* = 0 and $V_{rad}$ is relative radial speed of the UAV—speed of movement either toward or away from the target.
$$\{\begin{array}{c} V_{rad}<0,\;\mathrm{UAV}\;\mathrm{is}\;\mathrm{moving}\;\mathrm{away}\;\mathrm{from}\;\mathrm{the}\;\mathrm{target}\;\\ \;V_{rad}>0,\;\mathrm{UAV}\;\mathrm{is}\;\mathrm{approaching}\;\mathrm{the}\;\mathrm{target} \end{array}$$

The amplitude factor $K_{t}\;$ of the target echo can be written as:(6)$$K_{t}={\left[\frac{G²\lambda^{2}}{{\left(4\pi \right)}^{3}R_{t}^{4}L_{r}L_{a}}\cdot \sigma_{t}\right]}^{\frac{1}{2}}\cdot u\left(\eta -\frac{2R_{t}\left(t\right)}{c}\right)$$

The amplitude of the signal reflected by a potential target can be estimated according to Equation (6). It is a mathematical format relating the radar-target distance and the various factors of the detection chain such as wavelength (or frequency), antenna gain, radar loss, degradation of the signal at the level of the atmospheric propagation medium, and the most important element in the detection chain—reflectivity of the target $\sigma_{t}$ which varies for the same target according to the angle of view and the sea surface, the waves of which can partially or even completely hide the target.

### 2.2. Echo Signal Model of Sea Clutter

The radar does not only receive the signal reflected by an object of interest, indeed, the received radar echo is a combination of several signals reflected by obstacles in the maritime environment.

In various applications, in particular radar, electromagnetic scattering by a sea surface constitutes an important problem. To characterize this diffusion problem, there are different empirical or electromagnetic but also statistical (which are used for practical applications as is the case in this paper) models. In some conditions, signals detected by the receiver are the summation of propagated and reflected waves. The echo signal of sea clutter can be compounded as amplitude factor $K_{sea}\;$ and Doppler velocity factor. The signal can be written as:(7)$$S_{sea}\left(t\right)=K_{sea}\cdot A\left(t\right)\cdot e^{j\emptyset_{c}}$$

A(t) obeys the given statistical distribution. The radar clutter can be described by the random process [23,24,25,26,27]. Then, the amplitude factor of sea clutter is calculated by:(8)$$K_{sea}={\left[\frac{p_{t}G²\lambda^{2}}{{\left(4\pi \right)}^{3}R_{t}^{4}L_{r}L_{a}}\cdot \sigma_{c}\right]}^{\frac{1}{2}}$$

RCS or mean reflectivity of sea clutter, denoted as $\sigma_{c}$, is an important parameter for target detecting. It helps to describe the detectability of a marine target to radar, a high clutter RCS induces a less detectable target. In effect, RCS is a measure of the power of the return signal from an irradiated sea area by the radar. It is often beneficial to model the normalized RCS (NRCS), $\sigma_{0}$, that is:(9)$$\sigma_{c}=\sigma_{0}\cdot A_{c}$$
where $A_{c}$ is the illuminated area. 

## 3. Propagation Environment—Effect of Atmospheric Medium and Sea Surface on Radar Signal Characteristics

Before electromagnetic waves used for the detection function by the radar on board the drone reaches the sea surface, they must pass through the Earth’s atmosphere over a certain distance, more precisely the troposphere layer that extends from the earth surface up to approximately 20 km above it and includes climatic phenomena such as rain, snow, cloud, fog, wind, and storms. The traversed Earth’s atmosphere is bounded by the sea surface, which acts as an imperfect mirror. In general, the propagation channel (atmosphere and sea surface) affects radar signal in two ways: first, there is absorption, scattering, of electromagnetic energy by gases and water drops (in the form of rain or fog) which cause radar signal attenuation (loss). Second, there is a returned signal from the sea surface which ‘clutters’ the radar returns and can mask targets of interest.

This section deals with the influence of the atmosphere and the sea on low RCS targets detection by X-Ku bands radars located on a small drone flying at an altitude below 1000 m. Theoretical models allowing us to estimate the attenuation in the atmosphere, in particular the weather factors that interest us in this paper (gas, fog, and rain), will be presented, then the impact of the maritime surface surrounding the objects of interest will be discussed.

### 3.1. Atmospheric Precipitation Losses and Weather Environment Models

Figure 3 illustrates the geometry of the observation chain and the wave attenuation by the atmospheric gases, fog, and rain. Such weather factors have a serious impact on the path loss, which is directly related to the radar range. The further the radar wave and returning echo must travel through this medium, the greater the attenuation and the subsequent decrease in detection range. In X-Ku bands and above, radar signals passing through the Earth atmosphere are subjected to the following major effects: absorption, reflection, refraction, changes of polarization, scattering, and diffusion. Thus, these phenomena induce an upper limit on the frequency band used for any given application. Hence, it is necessary to have detailed attenuation models to accurately describe the two-way radar losses due to the atmospheric medium, especially in the case of harsh conditions.

A certain amount of attenuation takes place even when radar waves travel through a clear atmosphere. This effect is due primarily to the absorption resonance lines of oxygen and water vapor, with smaller contributions coming from nitrogen gas. The atmospheric gas attenuation in dB at frequencies up to 1000 GHz can be modeled based on ITU-R P.676-10 recommendation [28].

The specific gaseous attenuation is given by: (10)$$\gamma_{\mathrm{gas}}=\gamma_{0}+\gamma_{w}=0.1820\;f\;N^{\prime \prime }\left(f\right)\;\left(\mathrm{dB}/\mathrm{km}\right)$$

Here, $\gamma_{o}$ and $\gamma_{w}$ are the specific attenuations (dB/Km) due to dry air and water vapor, respectively, f is the frequency (GHz), and $N^{\prime \prime }\left(f\right)$ is imaginary part of frequency-dependent complex refractivity: (11)$$N^{\prime \prime }\left(f\right)=\underset{i}{\sum }S_{i}F_{i}+N_{D}^{\prime \prime }\left(f\right)$$

$N_{D}^{\prime \prime }\left(f\right)$ is the dry continuum due to pressure-induced nitrogen absorption and the Debye spectrum.

The line strength is given by: (12)$$S_{i}=a_{1}\times 10^{-7}p\;\vartheta^{3}\exp\left[\;a_{2}\left(1-\vartheta \right)\right]\;\;\mathrm{for}\;\mathrm{oxygen}\;=b_{1}\times 10^{-1}e\;\vartheta^{3.5}\exp\left[b_{2}\left(1-\vartheta \right)\right]\;\mathrm{for}\;\mathrm{water}\;\mathrm{vapor}$$

p and e are dry air pressure and water vapor partial pressures in hPa, and ϑ=$\;\frac{300}{T_{em}}\;$, where $T_{em}$ is the absolute temperature in K.

$F_{i}$ is the line-shape factor:(13)$$F_{i}=\frac{f}{f_{i}}\left[\;\frac{\Delta f-\delta \left(f_{i}-f\right)}{{\left(f_{i}-f\right)}^{2}+\Delta f^{2}}+\frac{\Delta f-\delta \left(f_{i}+f\right)}{{\left(f_{i}+f\right)}^{2}+\Delta f^{2}}\;\right]$$
where $f_{i}$ is the line frequency and Δf is the width of the line:(14)$$\Delta f=a_{3}\times 10^{-4}\left(p\;\vartheta^{\left(0.8-a_{4}\right)}+1.1\;e\;\vartheta \right)\;\;\mathrm{for}\;\mathrm{oxygen}\;=b_{3}\times 10^{-4}\left(p\;\vartheta^{b_{4}}+b_{5}\;e\;\vartheta^{b_{6}}\right)\;\mathrm{for}\;\mathrm{water}\;\mathrm{vapor}$$
with the coefficients $a_{1},a_{2},a_{3},a_{4},b_{1},b_{2},b_{3},b_{4},b_{5}\;$, and $b_{6}$ given in [28].

Thus, the gas attenuation according to the range $R_{t}$ can be defined as $L_{\mathrm{gas}}$ in Equation (15):(15)$$L_{\mathrm{gas}}=\gamma_{\mathrm{gas}}\cdot R_{t}\;\left(\mathrm{dB}\right)$$

In addition, cloud or fog and other forms of precipitation, including rain, can adversely affect the performance of radar equipment in two ways: by the introduction of signal power loss, thus reducing the maximum range of detection of a given target, and second, by producing an obscuration or masking of a given target by echoes from the precipitation itself.

Fog and cloud attenuation are the same atmospheric phenomenon. Fog is often characterized by the liquid water density. A medium fog with a visibility of roughly 300 m has a liquid water density of $0.05\;g/m^{3}$. For heavy fogs where the visibility drops to 50 m, the liquid water density is about 0.5 g/$m^{3}$. In most cases, clouds do not actually produce echoes on the radar, but a very dense fog bank can produce an attenuation of the radar signal, and therefore a significant reduction in detection range. This attenuation can be modeled based on the ITU model, Recommendation ITU-R P.840-6: Attenuation due to clouds and fog [29]. This model is valid for frequencies 10–1000 GHz.

The specific attenuation: (16)$$\gamma_{cloud}=k_{l}M\;\left(\mathrm{dB}/\mathrm{km}\right)$$

$k_{l}$ is specific attenuation coefficient $\left((\mathrm{dB}/\mathrm{km}\right)/\left(g/m^{3}\right))$ and M is liquid water density in the cloud or fog $\left(g/m^{3}\right)$.
(17)$$K_{l}=\frac{0.819\;f}{\varepsilon^{\prime \prime }\left(1+\eta_{l}^{2}\right)}$$
where *f* is the frequency (GHz), and: (18)$$\eta_{l}=\frac{2+\varepsilon^{\prime }}{\varepsilon^{\prime \prime }}\;$$

$\varepsilon^{\prime }$ and $\varepsilon^{\prime \prime }$ are the complex dielectric permittivity of water.

Then, cloud or fog attenuation according to the range $R_{t}$ can be written as: (19)$$L_{cloud}=R_{t}\cdot \gamma_{cloud}\;\left(\mathrm{dB}\right)$$

In the case of rain, the particles which affect the scattering and attenuation take the form of water droplets. Scattering is a physical process by which a raindrop in the path of an electromagnetic wave continuously attracts energy from the incident wave and reradiates that energy in all directions. Therefore, the raindrop may be thought of as the point source of the scattered energy. This contributes to the decrease of the maximum range of the detectability as discussed in [30,31].

The rain backscattering can be expressed by multiplying the rain reflectivity, $\eta_{r}$, by the resolution cell volume, *V,* as shown in Equation (20):(20)$$\sigma_{r}=\eta_{r}V$$

The rain reflectivity can be expressed as the summation of all raindrops per unit volume [32]:(21)$$\eta_{r}=\int_{0}^{\infty }\sigma \left(\lambda ,D\right)N_{D}\left(D\right)dD$$
where σ(λ,D) is the rain backscattering from a single small conducting sphere such as a raindrop, having diameter *D* < 0.2 λ (in the Rayleigh region, circumference ≪ wavelength), varies approximately as [33]:(22)$$\sigma \left(\lambda ,D\right)\sim \;9\times \frac{16\pi^{5}{\left(\frac{D}{2}\right)}^{6}}{\lambda^{4}}\sim \frac{688.5\;D^{6}}{\lambda^{4}}\;\left(m^{2}\right)$$

Mass, of course, depends on $D^{3}$, so scattering for a given mass of raindrops per cubic meter is very dependent on the statistical distribution of droplet diameters. The Marshall–Palmers drop size distribution (DSD) [34]:(23)$$N_{D}\left(D\right)=N_{0}\exp\left(\Lambda D\right)\;\left({\mathrm{cm}}^{-4}\right)$$

Here, $N_{0}=0.08\;{\mathrm{cm}}^{-4}$, $\Lambda =41\rho^{-0.21}{\mathrm{cm}}^{-1}$ with ρ is rain rate (mm/h) and *D* is the drop size diameter in cm.

The amount of attenuation in a region of rainfall can be related to the rate of precipitation. According to [35], the rain rate can range from less than 0.25 mm/h for very light rain to over 50 mm/h for extreme rains. If the size of the water droplet is an appreciable proportion of the 3 cm wavelength, then strong clutter echoes are produced followed by a severe loss of energy. Rainfall attenuation is computed according to the ITU rainfall model, Recommendation ITU-R P.838-3: Specific attenuation model for rain used in prediction methods [36]. The specific attenuation $\gamma_{rain}$ (dB/km) for frequencies from 1–1000 GHz is modeled as a power law relationship with respect to rain rate ρ (mm/h):(24)$$\gamma_{rain}=k\rho^{\alpha }\;\left(\mathrm{dB}/\mathrm{km}\right)$$

Values for the coefficients k and α are determined as functions of frequency, f(GHz).
(25)$$\log_{10}k=\overset{4}{\underset{j=1}{\sum }}\left(a_{j}\;\exp\left[-{\left(\frac{\log_{10}f-b_{j}}{c_{j}}\right)}^{2}\right]\right)+m_{k}\log_{10}f+c_{k}$$
(26)$$\alpha =\overset{5}{\underset{j=1}{\sum }}\left(a_{j\;}\exp\left[-{\left(\frac{\log_{10}f-b_{j}}{c_{j}}\right)}^{2}\right]\right)+m_{\alpha }\log_{10}f+c_{\alpha }$$
with the coefficients $a_{j},b_{j},c_{j},m_{k},c_{k},m_{\alpha }$, and $c_{\alpha }$ given in [36].

Finally, the rainfall attenuation in terms of the range $R_{t}$ and the scale factor of *r* can be defined as: (27)$$L_{rain}=rR_{t}\gamma_{rain}\;\left(\mathrm{dB}\right)\;$$
(28)$$r=\frac{1}{0.477R_{t}{}^{0.633}\rho_{0.01}^{0.073\alpha }f^{0.123}-10.579\left(1-\exp\left[-0.024\;R_{t}\right]\right)}$$

Table 2 gives representative values for gas, cloud, and rain attenuations at X-Ku bands of frequencies, customarily used in maritime airborne radars.

As shown in Table 2, radar signal loss in the atmosphere is influenced by numerous factors. It increases with both frequency and radar/target range. The loss computed with the rain model is mostly larger than the losses computed with gas model or fog model. Therefore, the greatest adverse effect is mainly dependent on rain rate which leads to serious reductions of detection range. In addition to the radar frequency, the effect of rain is also dependent on several parameters such as radar polarization, pulse duration, and beamwidth. The features shown in Table 3 indicate the reduction in the rain clutter returns as function of elevation and azimuth beamwidths.

During its way to a target and back to the radar, total attenuation of radar signal through unobstructed atmosphere at any frequency is the sum of attenuation caused by oxygen absorption and attenuation caused by water vapor absorption. Cloud and rain attenuation, when present, adds an additional element.

Two-way attenuation (dB): (29)$$L_{a}=2\times \left(L_{\mathrm{gas}}+L_{cloud}+L_{rain}\right)$$

Based on above atmospheric recommendations and Equation (29), Figure 4 calculates the radar signal attenuation versus detection range in different weather conditions. The atmosphere is lossy, particularly during precipitation. However, the atmospheric effect on the received echo did not always obey Equation (29) as described. In some cases, the signal loss may be stronger than predicted by this equation. The reason for this uncertainty is dissipation due to refraction (lensing) [37], which is not considered in this equation.

When the terms are all expressed in dB or dBW, as shown in the Equation (30), we handle the above attenuations by inserting an atmospheric loss term, $L_{a}$, in the equation of amplitude factor of target echo, and its measurements and simulation according to the target range $R_{t}$ with different precipitation characteristics compared to the amplitude factor of sea clutter are presented in Section 3.2.
(30)$$K_{t}=136.56+p_{t}+2\;G+\sigma_{t}-20\;\log_{10}\left(f\right)-40\log_{10}(R_{t})-L_{r}-2\left(\;L_{\mathrm{gas}}+L_{cloud}+L_{rain}\right)$$

### 3.2. Sea Scatter and Its Impact on Targets Detection

Although the atmosphere may contain various particles that degrade the radar signal strength and the reduction in the radar range which results therefrom, in this environment, the main limitation that impacts the radar detection capability is radar sea clutter, which is usually associated with particular characteristics of the sea surface. In contrast with ground surface, the sea is a dynamic system with two-dimensional propagation. This dynamic is characterized by the sea state (Table 4) and the direction of the sea waves generated by two different mechanisms: the capillarity waves and the gravity waves [38]. The first kind of waves are of small amplitude and in very large number, having a short wavelength while the gravity waves have larger amplitudes and larger wavelengths.

#### 3.2.1. Empirical Sea Clutter Model for Airborne Radar

The sea surface physique nature is an important parameter in determining the strength of the scattered signals back to the radar. It is always changing, and its complex nature implies that the sea clutter RCS fluctuates widely around the mean value as determined by the NRCS $\sigma_{0}$, which has been the subject of numerous calculations, depending on the sea state, the radar look direction, wavelength, and polarization. In this sense, researchers have published several empirical models. Examples of such models include the TSC (Technology Service Corporation, Arlington, VA, USA) model [39], the GIT (Georgia Institute of Technology, Atlanta, GA, USA) model [23], Hybrid model [40], NRL model [41], and a model developed by the Royal Radar Establishment (RRE) in United Kingdom [24].

However, these models do not always agree with the most complete experimental database of sea clutter reflectivity. They are therefore only approximate models. Nonetheless, the GIT model has widespread acceptance as an appropriate model for the mean radar cross section. This model has undergone a long period of development and now spans a frequency range of 1–100 GHz, wave heights up to 3.5 m, all wind directions, and both HH and VV polarizations, but only covers grazing angles below 10°. Therefore, it was found to be suitable for low airborne maritime radar data, which typically operate at low grazing angles (<10°), either from necessity or to achieve the best detection performance against low RCS targets.

For a surface area illuminated by the radar resolution cell, the clutter RCS is determined by multiplying the illuminated area $A_{c}$ by the mean clutter reflectivity $\sigma_{0}$ which is dimensionless and is often expressed by a negative dB value (a larger reflectivity leads to a smaller negative dB value). The clutter RCS in the resolution cell is given by:(31)$$\sigma_{c}=\sigma_{0}\cdot \;\alpha_{f}R_{t}\theta_{az}\rho \sec\left(\varphi \right)\;$$

Here, *R**_t_* is the distance between radar and sea clutter area element, $\theta_{az}$ is the radar beam width of 3 dB, in radian form, and φ is the grazing angle. The range resolution ρ, is related to the radar pulse bandwidth, B by $\rho =\frac{c}{2B}$. The factor $\alpha_{f}$ accounts for the actual compressed pulse shape and the azimuth beamshape, including the range and azimuth sidelobes. For a rectangular-shaped pulse and beamshape, $\alpha_{f}=1$, while for a Gaussian-shaped beam and rectangular pulse, $\alpha_{f}=0.753$. 

Different sea clutter RCSs are computed and plotted in Figure 5, and the distance range is from 1 km to 25 km (short/medium range radar). The RCS of sea clutter will decrease when the distance increases. For calm sea surfaces, usually the sea state is equal to 1. Even in a very near range of 1 km, 10 GHz, and horizontal polarization, the RCS of the area element is less than −30 dBsm or 0.001 m². At low sea state, where gravity waves are small, the individual capillaries are to some degree organized or correlated so their RCS statistical distributions shown some departure from the Gaussianity. Sea clutter is often therefore processed as an ordinary noise. A sea state of 6 is typical for active seas. Sea clutter RCS in this state is nearly 5 dBsm or 3.16 m² in 1 km for vertical polarization, and −6.4 dBsm in 25 km. At high sea state, quite apart from abnormal waves, local collisions between individual waves in confused seas wavelengths throw up transient sea spikes, a few meters high, a few meters wide, lasting a few seconds, and moving at nearly wind velocity.

We handle the above sea reflectivity by insertion of the normalized radar cross section term, $\sigma_{0}$, and the area of illumination, $A_{c},$ in the amplitude factor of sea clutter equation.
(32)$$K_{sea}=136.56+p_{t}+2\;G+\sigma_{0}+10\cdot \log_{10}(A_{c})-20\cdot \log_{10}\left(f\right)\;\;40\cdot \log_{10}(R_{t})-L_{r}-2\;\left(\;L_{\mathrm{gas}}+L_{cloud}+\;L_{rain}\right)\;\;$$

The sea clutter power in a resolution cell is proportional to its area, improving range resolution will reduce it. Figure 6 shows the interest of high resolution (low ρ value) to increase contrast between the target echo and the surrounding sea clutter echoes. The amplitude factor of sea clutter has a generally sinusoidal variation with wind direction, with peak in the upwind direction (0°), while its minimum value is calculated in downwind direction (±180°). 

Based on Equations (30) and (32), Figure 7 shows simulation and comparison between the amplitude factors of a target with RCS = 1 m² and sea clutter versus detection range. The solid line shows the simulation under normal weather conditions, whereas the dashed line is the simulation of amplitude factors in abnormal atmospheric conditions including water vapor density = $25\;g/m^{3}$, liquid water density = $0.5\;g/m^{3}$, and rainfall = 16 mm/h. Not only does the returned signal strength decrease with range, but it is also weaker in the case of abnormal atmosphere.

#### 3.2.2. Sea Clutter Amplitude with Compound K-Distribution Model

The clutter also fluctuates in distributed amplitude as large spikes that are represented by its statistical distribution or probability density function (PDF). This results in false alarms and makes detection more difficult by traditional constant false alarm rate (CFAR) technologies [20,21,22]. Models for the amplitude distribution of sea clutter are usually developed empirically from measurements of real data as it is not currently possible to accurately predict the PDF of sea clutter under different conditions using physical models of the sea surface. The most popular models for describing sea clutter amplitude distributions are Rayleigh [23], Log-normal [25], and Weibull [26], but the compound k distribution [23,24,27] has become perhaps the most popular one. This model was developed to describe the amplitude statistics of sea clutter observed with airborne maritime survey radars with a range resolution very much less than the sea swell wavelength. When the high-resolution radar observes the sea surface at a small grazing angle, the clutter distribution presents a significant non-Gaussian characteristic [42]. Since an airborne forward-looking scanning radar is used for detection with small grazing angle, it is reasonable to model the sea clutter amplitude as K-distribution, which can be expressed as:(33)$$p\left(x\right)=\frac{2\cdot b}{\Gamma \left(v\right)}\;\cdot {\left(\frac{b\cdot x}{2}\right)}^{v}\cdot k_{v-1}\left(b\cdot x\right)\;\;$$
where v and b are the shape parameter and the scale parameter for the K-distribution, respectively. $k_{v-1}\left(b\cdot x\right)$ is the modified Bessel function of the second kind Γ(·) is the Gamma function.

Figure 8a demonstrates the compound K-distribution (CKD) amplitude model PDF in Equation (33) for *b* = 1 and varying shape parameters, 0.3, 1, 3, and 10.

The CKD amplitude model is characterized by a scale parameter, b, and a shape parameter, v, which depend on sea conditions and the radar parameters. The first represents the power characteristics of the returned signal. The smaller the value of b, the more powerful the reflected signal is from the sea surface, and this can be predicted from the knowledge of the mean clutter reflectivity and the radar parameters. The second parameter provides a measure of the amplitude spikiness; the smaller the value of the shape parameter, the spikier the sea clutter amplitude. It can be determined by the empirical model [43] that can be used for grazing angles in the range 0.1° to 10°.
(34)$$\log_{10}\left(v\right)=\frac{2}{3}\log_{10}\left(\varphi \right)+\frac{5}{8}\log_{10}\left(A_{c}\right)-k_{pol}-\frac{\cos\left(2\theta_{s}\right)}{3}\;$$
where $A_{c}$ is the radar resolved area, $k_{pol}$ is the radar transmit and receive polarization parameter, taken to be 1.39 for vertical polarization (VV) and 2.09 for horizontal polarization (HH), and $\theta_{s}$ is the aspect angle with respect to swell direction, which can be omitted when there is no swell. Under these conditions, the spikiness of sea clutter decreases with grazing angle, φ, and increases in the upwind and downwind directions, and horizontal polarization is much less effective [44,45].

#### 3.2.3. Signal-to-Clutter Ratio

Radar returns from the sea surface interfere with the desired target. Modeling the detection performance of a target in clutter requires models for the power of the target $p_{r}$, clutter $p_{c}$, and noise $p_{n}$.

Due to a larger contrast between the power of sea clutter and the power of noise, clutter-to-noise ratio (CNR) is often expressed by a positive dB value. Backscattering from the sea is then the most disturbing unwanted signal, its power makes the detection function quite difficult in this environment. The interference level can be characterized by the ratio of the received signal power ($p_{r}$) to the clutter power ($p_{c}$), i.e., ($p_{r}$/$p_{c}$), which must be used in place of ($p_{r}$/$p_{n}$) in calculating radar detection and measurement performance in maritime environment. Thus, the SCR can be defined as:(35)$$\mathrm{SCR}=10\cdot \log_{10}\left(\frac{p_{r}}{p_{c}}\right)\;\left(\mathrm{dB}\right)\;$$

We can also consider, in the case of pulse radar, that the target and the clutter are confused only if they are at the same distance and directly compare the equivalent surfaces of target and clutter.

Signal-to-clutter ratio is: (36)$$\mathrm{SCR}=10\cdot \log_{10}\left(\frac{\sigma_{t}}{\sigma_{c}}\right)=10\cdot \log_{10}\left(\frac{\sigma_{t}\cdot \cos\left(\varphi \right)}{\alpha_{f}\cdot \sigma_{0}\cdot \rho \cdot \theta_{az}\cdot R_{t}}\right)$$

The SCR of a detection cell can obtain significant values that give satisfactory detection performance for surface target with large RCS. However, several of the diverse types of ships now at sea, including some high-speed craft, have poor RCS by reason of small size, streamlined shape, and sometimes non-metallic construction. The RCS of practical targets such as ships is difficult to measure and more difficult to calculate with any degree of precision. The measurement process is based on the standard techniques used for measuring the RCS of ships [46,47,48,49]. The relationship of RCS with the size of a wide range of extended maritime targets is discussed in the Figure 9 [50]. 

By considering a small object (1 m²) and a medium object (6 m²), Figure 10 shows various values of signal-to-clutter ratio of these objects in different sea states and at different object distances, while, in order to analyze and consider the sea clutter in the problem treated, Figure 11 simulates the distribution model K composed of the sea clutter in the case of different sea states and in the presence of targets with different values of RCS. According to the simulation, the following remarks can be made:SCR increases with the distance separating the radar and the resolution cell containing the target. Detection of a target of the same RCS is more difficult in the case where this target is close to the radar, because as we have seen, the clutter RCS decreases with distance.The smaller the reflectivity $\sigma_{t}$ of the target to be detected, the weaker the SCR. Weaker target responses, as from small vessels, will be undetectable when their echoes are not stronger than that of the sea clutter. Therefore, when clutter is severe, a high RCS is necessary.SCR decreases with sea state. Thus, in a calm sea state, the influence of clutter is weak, the echo only has to compete with received sea clutter, and quite low RCS may give adequate detection range. However, in a rough sea state, surface backscattering covers the object echo that is submerged by the clutter and influences its detection.

**Figure 10 sensors-22-03263-f010:**
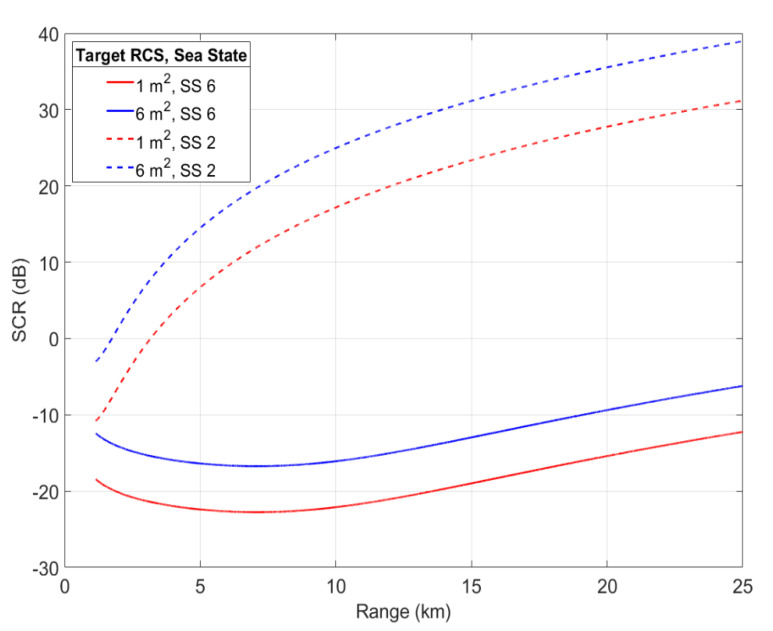
Simulation of signal-to-clutter ratio versus detection range.

**Figure 11 sensors-22-03263-f011:**
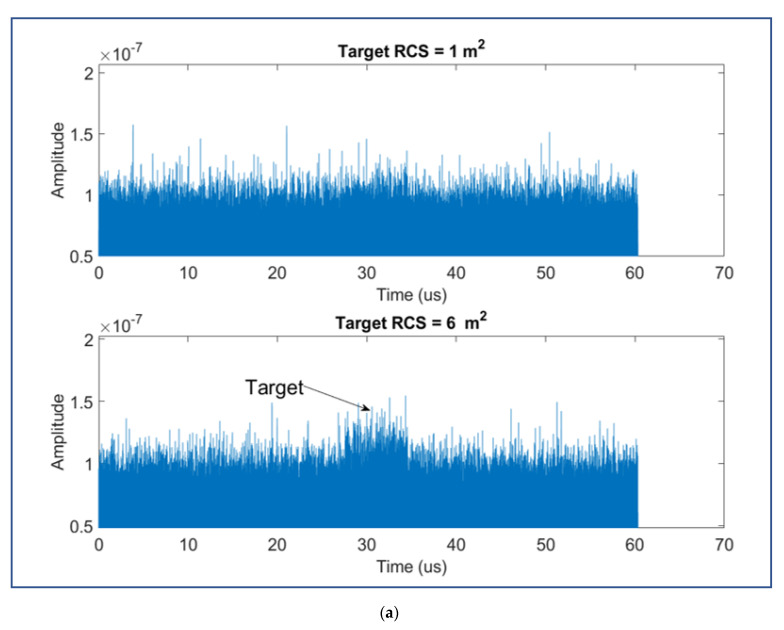
(**a**) impact of different object RCSs; (**b**) impact of different sea states.

## 4. Doppler Frequency of Target Based on Scenario of UAV Dynamic Model

Targets can only be detected and tracked when the echo power or signal can be distinguished or detected with a reasonable certainty from competing clutter. Many radars only use the received signal envelope in their signal processing. The target presence decision, or detection decision, is made on a set of detection cells splitting the detection range, as shown in Figure 12b. The total number of detection cells can be calculated according to the values in radians of the angle covered by the antenna scan, the lobe at −3 dB in azimuth $\theta_{az}$, the distances maximum and minimum instrumented $D_{MAX}$ and $D_{MIN}$, respectively, and the radial extension of the detection cell; this value can often correspond to a distance sampling step corresponding to the range resolution ρ: (37)$$N_{-}det=\left(\frac{\pi }{\theta_{az}}\right)\left(\frac{D_{MAX}-D_{MIN}}{\rho }\right)\;$$

Non-coherent statistics may be described in terms of either the envelope (linear detector) or the power (square law detector). For the detection probability to be maximized, the power backscattered by a boat in a detection cell must be greater than the power backscattered from the adjoining cells, which are assumed to contain only the sea clutter. The clutter power in a resolution cell is proportional to its area, this then requires incoherent detection on high resolution radar to reduce the area of the detection cell. In this situation, if the RCS of target to be detected is not too low and under normal weather and sea conditions, then traditional CFAR technologies can effectively detect the object echo in the compound radar echo, which also includes sea clutter echo and thermal noise. Then, the object can be tracked by estimating the trajectory through the tracking filter [51,52]. The search for high resolution is, however, limited; when the surface area of the boats can no longer be contained in a single cell, RCS is spread over several cells. In such cases, it is no use improving range resolution; contrast enhancement in relation to sea clutter is very small. As we have seen in the previous part, sea clutter featuring as a lot of spikes will decrease the target detection performance and the electromagnetic power of certain objects can be drowned in the sea clutter, especially in rough states. This method can obtain satisfactory detection performance for a surface target with a large RCS. Consider that the existence of small moving target may not present large amplitude in the detection cell but may change the Doppler spectrum shape. For most cases, coherent integration is more efficient than non-coherent integration. Therefore, some radars specialized in the detection of small, fast-moving boats can use a Doppler mode with low-range resolution (75 to 150 m), even if the boats have low RCS. Using a higher dimensional detection variable, the Doppler spectrum, rather than the one-dimensional amplitude information, is expected to achieve improved detection performance. Coherent radars employing Doppler processing can distinguish target from clutter when the target radial speed is sufficiently high [53]. However, some targets of interest will have Doppler shifts that are not significantly different from the Doppler spectrum of the sea clutter. The radial speed of an object does not only depend on the speed of the radar platform and the object, but also on the view geometry (UAV dynamic model and scanning radar model). 

Thus, the aim of the rest of this section is to study the variation of the Doppler frequency, $f_{D},$ of an object intercepted by the airborne radar at 10 GHz according to the parameters represented in Figure 12 and Figure 13 including UAV speed, object speed, azimuth angle, and grazing angle: $f_{D}\equiv \left\{V_{d},V_{t},\;\theta ,\varphi \right\}$.

### 4.1. Geometry of the Problem: UAV Radar Measurement Model

In this section, the small UAV dynamic model and the airborne forward-looking scanning radar measurement model are given.

Figure 12 illustrates the geometry of the airborne forward-looking radar. We assume that the small UAV platform speed is $V_{d}$ and its altitude is H < 1000 m. The *y*-axis is parallel to the direction of $V_{d}$, and the x-axis is perpendicular to the *y*-axis. A target of interest is located on the sea surface, at a slant range $R_{t}\left(t\right)\cdot$ It can move in plane (xoy) at a speed $V_{tx}$ and $V_{ty}$, respectively, along the *x*-axis and the *y*-axis, and it is also located by two angles: the azimuth angle θ (positive to the right of the flight path, and negative to its left) and the grazing angle *φ* which is related to height of UAV and its distance from target, $\varphi =\arcsin\left(\frac{H}{R_{t}}\right)$. 

As the UAV flies, the antenna continuously scanned periodically to achieve a wide surveillance range. The scanning speed of radar beam is w, the azimuth angle varies during each sweep period between −$\;\frac{\pi }{2}$ and $\frac{\pi }{2}$.

### 4.2. Doppler Frequency of Target

Doppler is the apparent change in wavelength (or frequency) of radar-emitted signal due to relative movement between the UAV and the target. For a target traveling toward the radar, the frequency of the received signal exceeds the transmit signal frequency and the Doppler shift is positive. On the other hand, if a target is moving away from the radar, then received signal frequency will be smaller than the transmit signal frequency and the Doppler shift is negative. 

Assuming that the speed of UAV is greater than the speed of the target, in slow time t, the coordinate of UAV is set as (0,$V_{d}t,H)$. The position of the moving target is *(*$x_{0}+V_{tx}t,\;y_{0}+V_{ty}t)$, where *(*$x_{0},y_{0})$ is the initial position when t = 0. Instantaneous distance between the moving target and the radar platform can be expressed as:(38)$$R_{t}\left(t\right)=\sqrt{{\left(x_{0}+V_{tx}t\right)}^{2}+{\left(y_{0}+V_{ty}t-V_{d}t\right)}^{2}+H²}$$

As shown in Figure 13, a boat on the sea surface can be located according to its azimuth angle θ, its grazing angle φ, its distance $R_{t}$, and its radial speed $V_{rad}$. The radial speed of such a target is directly obtained from its speed itself $V_{t}$ and that of the drone $V_{d}$ and the angle γ (angle between direction of the radial speed vector and direction of the relative speed vector of the UAV to the target).
(39)$$V_{rad}=V_{d/t}\cdot \cos\left(\gamma \right)$$
where $V_{d/t}$ is relative speed of the UAV to the target (Figure 14):(40)$$\begin{array}{l} \vec{V_{d/t}}=\vec{V_{d}}-\vec{V_{t}}=\vec{V_{d}}-\vec{V_{ty}}-\vec{V_{tx}}\\ V_{d/t}=\sqrt{{\left(V_{tx}\right)}^{2}+\left(V_{d}-V_{ty}\right)²} \end{array}$$

The angle γ can be broken down into an elevation φ (grazing angle) and an azimuth θ. Then, the radial speed can be written as:(41)$$V_{rad}=V_{d/t}\cdot \cos\left(\varphi \right)\cdot \cos\left(\theta -\beta \right)\;=V_{d/t}\cdot \cos\left(\varphi \right)\cdot \left[\cos\left(\theta \right)\cdot \cos\left(\beta \right)+\sin\left(\theta \right)\cdot \sin\left(\beta \right)\right]\;V_{rad}=V_{d/t}\cdot \cos\left(\varphi \right)\cdot \left[\cos\left(\theta \right)\cdot \sqrt{1-{\left(\frac{V_{tx}}{V_{d/t}}\right)}^{2}}-\frac{V_{tx}}{V_{d/t}}\cdot \sin\left(\theta \right)\right]\;$$

Positive relative radial speed indicates that the radar platform is approaching the target. Negative relative radial speed indicates that the radar platform is moving away from the target.

The Doppler frequency $f_{D}$ of the target is retrieved by using the relative radial speed $V_{rad}$:(42)$$V_{rad}=\frac{\lambda f_{D}}{2}\;\to \;f_{D}=\frac{2\;V_{rad}}{\lambda }$$
(43)$$f_{D}=\frac{2\;}{\lambda }V_{d/t}\cdot \cos\left(\varphi \right)\cdot \left[\cos\left(\theta \right)\cdot \sqrt{1-{\left(\frac{V_{tx}}{V_{d/t}}\right)}^{2}}-\frac{V_{tx}}{V_{d/t}}\cdot \sin\left(\theta \right)\right]$$

Equation (43) is a mathematical formula for the Doppler frequency of the target intercepted by the radar located on the UAV platform with a dynamic model as described above.

In Figure 15, the Doppler frequency of the boat was simulated from Equation (43) for different azimuth angles $\theta \epsilon \left[-\frac{\pi }{2},\;\frac{\pi }{2}\right]$ (angle swept by the antenna) and for different UAV speeds.

In case (a), we consider that the boat is fixed, and we notice that its Doppler frequency is symmetrical with respect to the flight direction, then the fixed targets located symmetrically with respect to the flight path have the same Doppler frequency. In (b), we consider that the object is moving with a two-dimensional movement $\left(V_{tx},V_{ty}\right)=\left(-6\;,\;8\right)\;m/s$ in plan (xoy), and in this case, the Doppler frequency of the object is not symmetrical, it varies with the azimuthal position of the target. For (c) and (d), the boat motion is one-dimensional, it moves along the y-axis ($V_{tx}=0$ m/s and $V_{t}=V_{ty}$), except in case (c) where the boat speed sense is the same as that of the UAV, while in case (d), the boat is moving in the opposite sense to that of the UAV’s flight. For both cases, the Doppler frequency is symmetrical, with significant values in (d).

Thus, for a given grazing angle, φ, and a certain speed of the UAV which carries a radar, the antenna of which scans an azimuth angle between −$\;\frac{\pi }{2}$ and $\frac{\pi }{2}$, the Doppler frequency of a target on the scanned terrain varies depending on the module, the direction and the sense of both drone and target, and localization of the target in azimuth and in elevation. In fact, frequency offset can be significant when the radar is forward-looking (θ=0) and both the UAV speed vector and the target speed vector have the same direction but opposite senses. Thus, significant Doppler frequency can be defined as $f_{\mathrm{Dmax}}$ in Equation (44), and its variations are computed and shown in Figure 16:(44)$$f_{Dmax}=\frac{2}{\lambda }\cdot \left(V_{d}-V_{t}\right)\cdot \cos\left(\varphi \right)$$

Geometry corresponding to significative Doppler frequency is shown in Figure 17a. In contrast, it is minimum when the radar is side-looking ($\theta =\pm \frac{\pi }{2}$) and the target moves from away the UAV in the *x*-axis ($V_{ty}=0$ m/s and $V_{t}=V_{tx}$); in this case, the minimum Doppler depends only on target speed as shown in Figure 17b,c:(45)$$\;f_{Dmin}=\{\begin{array}{c} -\frac{2}{\lambda }\cdot V_{t}\cdot \cos\left(\varphi \right)\;if\;V_{t}>0\\ +\frac{2}{\lambda }\cdot V_{t}\cdot \cos\left(\varphi \right)\;if\;V_{t}<0 \end{array}$$

## 5. Conclusions and Future Work

In this article, we discussed the problems related to the detection chain (via a radar located on an aircraft) of small targets moving on the sea surface. In inclement weather, the ability of a radar to detect targets is decreased because of attenuation of the radar signal in the path between radar and target, and because of clutter from sea waves. In this context, we have particularly dealt with different points, and more specifically, the propagation channel consisting of the atmospheric propagation medium and the surface of the sea, as well as the detection geometry consisting of UAV dynamic model and scanning radar model. Low RCS targets detection and tracking in maritime environment is a traditional problem for maritime survey radars. In order to present the difficulties associated with this problem, we analyzed the maritime target detection performance of airborne radars located on a small UAV. Firstly, weather factor effects on the radar EM wave propagation were presented. Recommendations of atmospheric loss model to account for the attenuation of air, water vapor, and precipitation in the radar-target path were adopted. The effects of various meteorological conditions on the detection range capabilities radar were discussed, and a general equation of two-way path loss in the troposphere for the radar system was derived that can be used to compare the ability of the radar to detect targets in dry air with its ability to detect similar targets in fog and rain. Then, the impact of sea clutter to the radar detecting performance and the impact of several radar parameters (frequency, polarization, range resolution, etc.), different sea states, and different object distances was examined. Finally, Doppler frequency of an object according to its movement in the observation plane and radar view geometry from the UAV platform was analyzed. Simulation results show that serious reductions of detection range can occur, together with the masking of targets, in heavy rain and rough sea states, particularly at wavelengths of about 3 cm and shorter. These harsh weather factors do not benefit object detection and tracking due to the additional atmospheric loss in radar range equation and low signal-to-clutter ratio (SCR) of the radar echo. The Doppler frequency of a target can be an exploitable element to distinguish the moving target, especially when the target moves in the opposite sense but in the same direction of the flight of the platform, since in this case, the Doppler frequency takes its maximum value. This work will be continued on different aspects at the same time at the level of the sensor in order to effectively choose the various specific parameters (waveform, polarization, frequency, etc.) fitted to the problem in question. In addition, future works have to be carried out on methods for eliminating sea clutter. In addition, different detection algorithms are being developed with tests carried out in different situations. Moreover, partial measurements are planned in order to evaluate different aspects of the methodology developed and applied in this paper. Finally, the problem addressed in this paper is of significant interest in other applications, in particular the detection of various objects presents on the sea surface including floating objects from a ship equipped with various observation sensors.

## Figures and Tables

**Figure 1 sensors-22-03263-f001:**
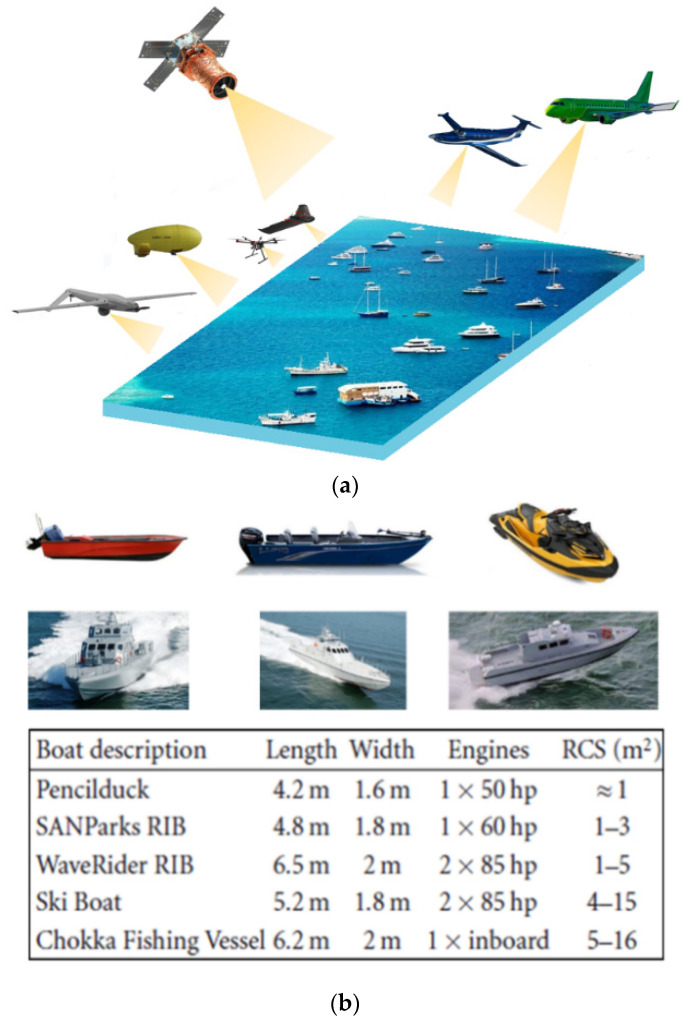
(**a**) An illustration of satellite, manned aircraft, and UAV remote sensing platforms; (**b**) small and medium maritime vessels.

**Figure 2 sensors-22-03263-f002:**
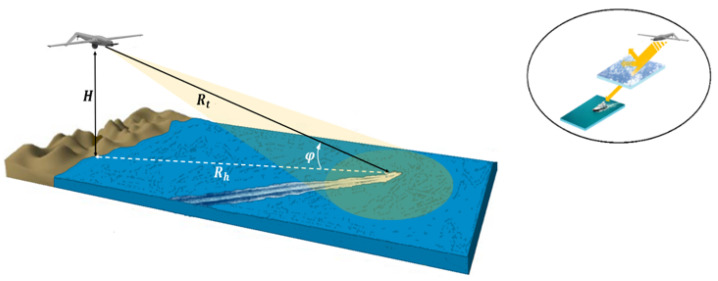
Forward-looking maritime radar located on UAV.

**Figure 3 sensors-22-03263-f003:**
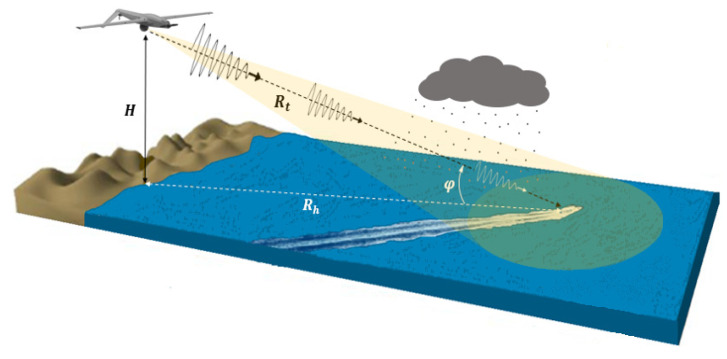
EM wave propagation attenuation by atmospheric factor.

**Figure 4 sensors-22-03263-f004:**
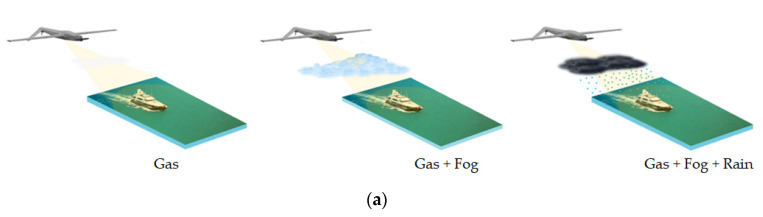
(**a**) Marine observation in different weathers; (**b**) radar signal attenuation versus detection range.

**Figure 5 sensors-22-03263-f005:**
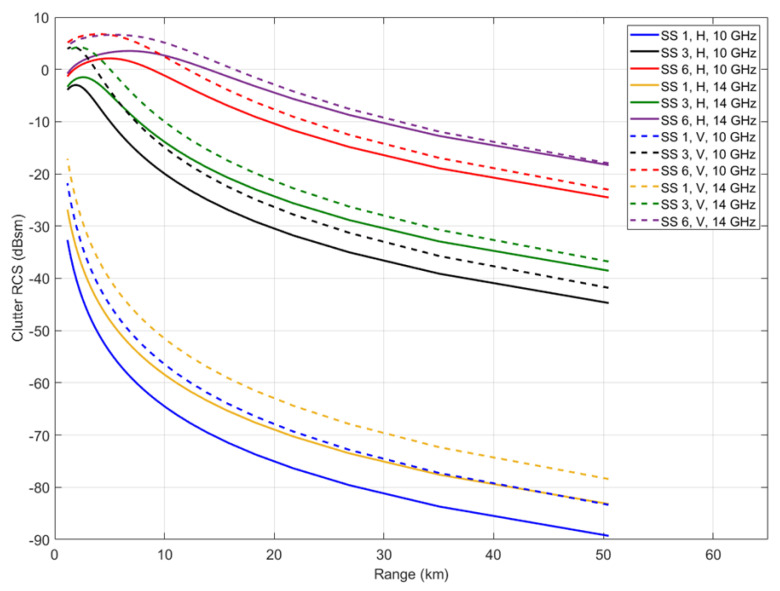
Typical radar cross-section of clutter versus range, frequency, polarization, and sea-state (SS).

**Figure 6 sensors-22-03263-f006:**
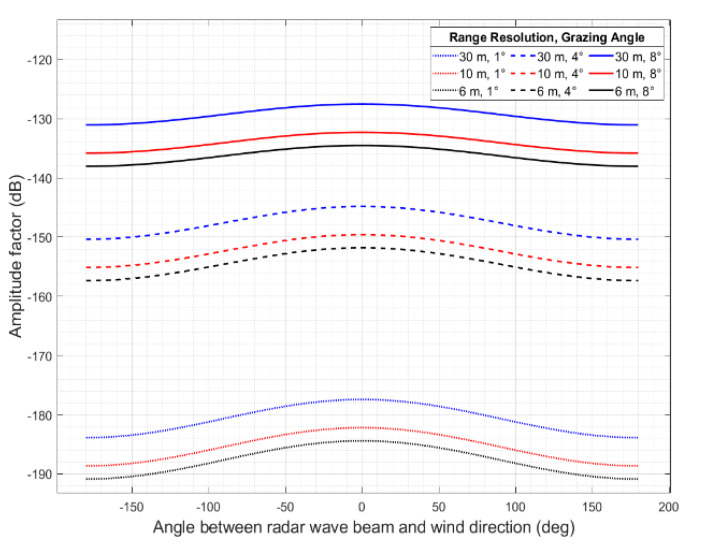
Amplitude factor of sea clutter.

**Figure 7 sensors-22-03263-f007:**
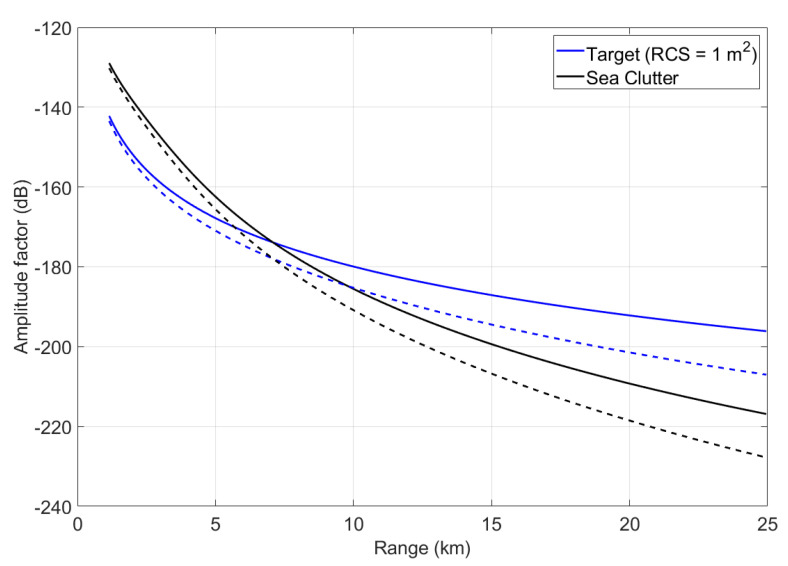
Amplitude factors versus detection range.

**Figure 8 sensors-22-03263-f008:**
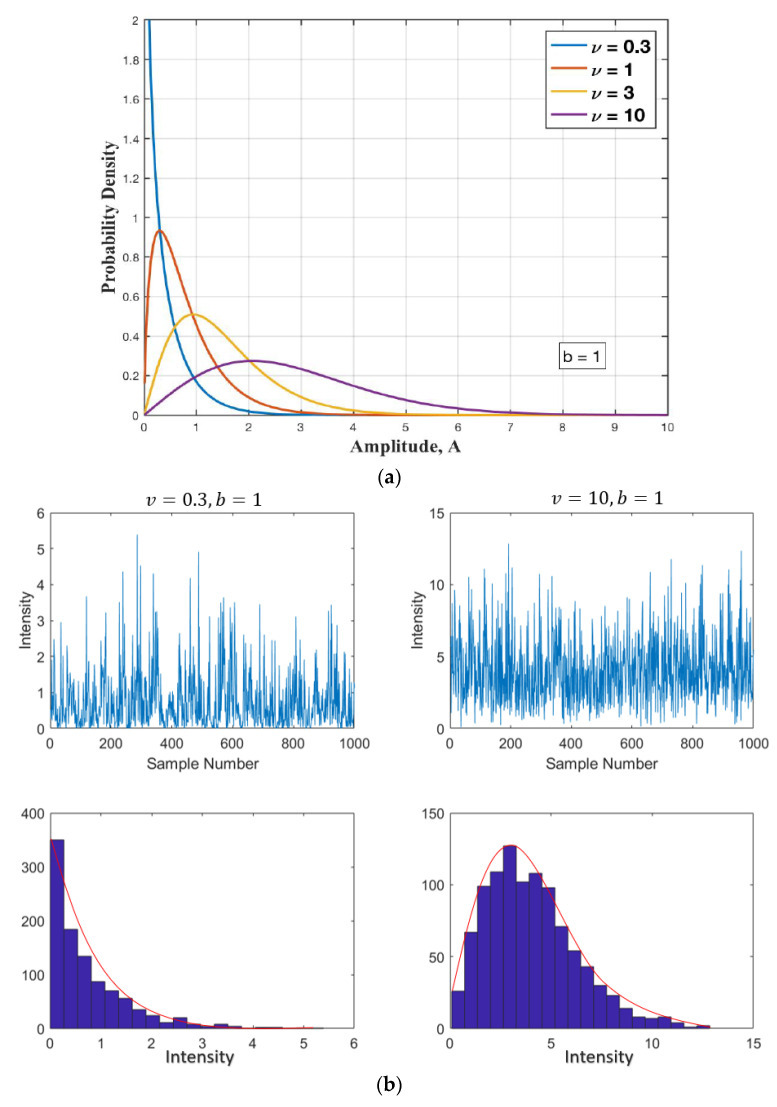
(**a**) CKD amplitude model PDF in Equation (33) for scale parameter *b* = 1 and for varying v; (**b**) simulation of K-distribution random variates for different values of shape parameter v.

**Figure 9 sensors-22-03263-f009:**
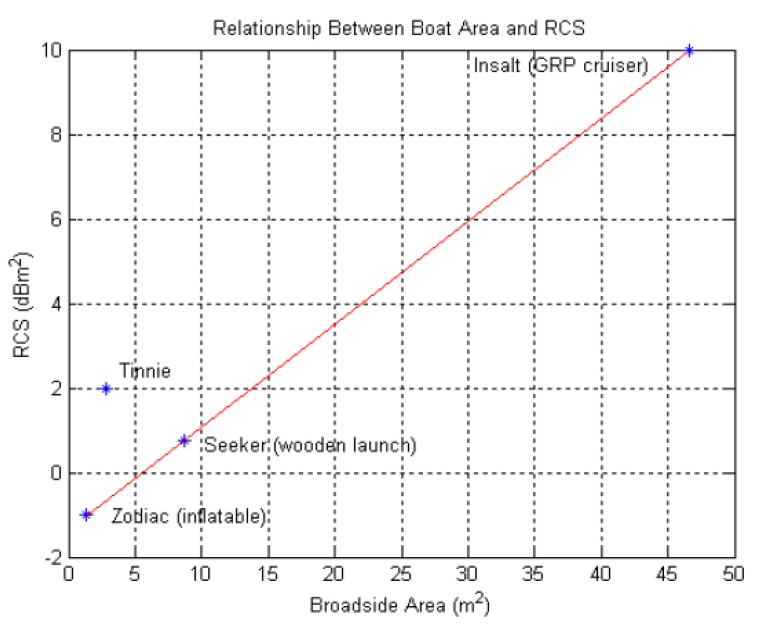
Relationship between the boat projected area and the median RCS [50].

**Figure 12 sensors-22-03263-f012:**
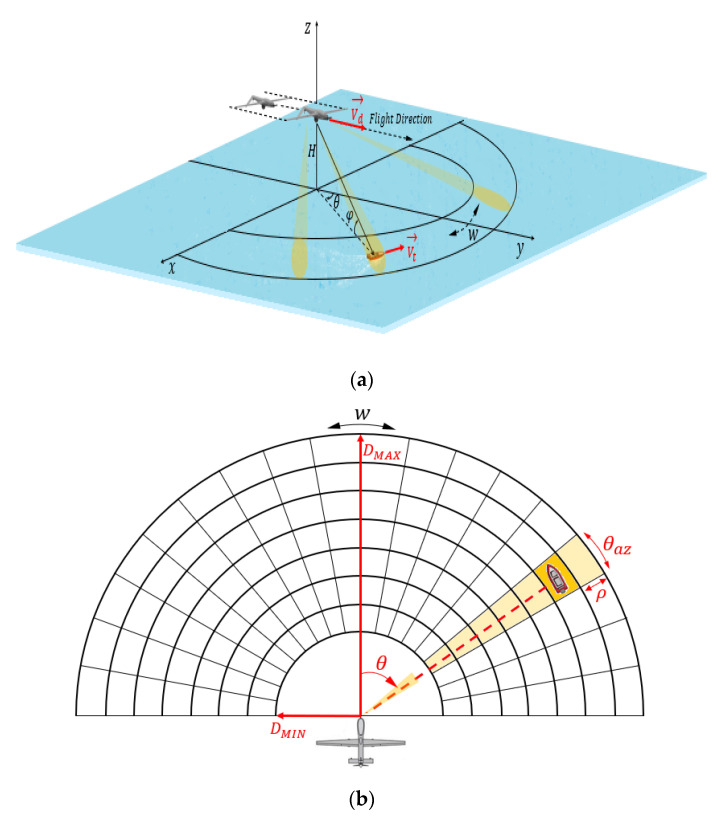
Geometrical configuration of the radar and a target: (**a**) 3D geometry of radar; (**b**) periodic scanning of radar antenna.

**Figure 13 sensors-22-03263-f013:**
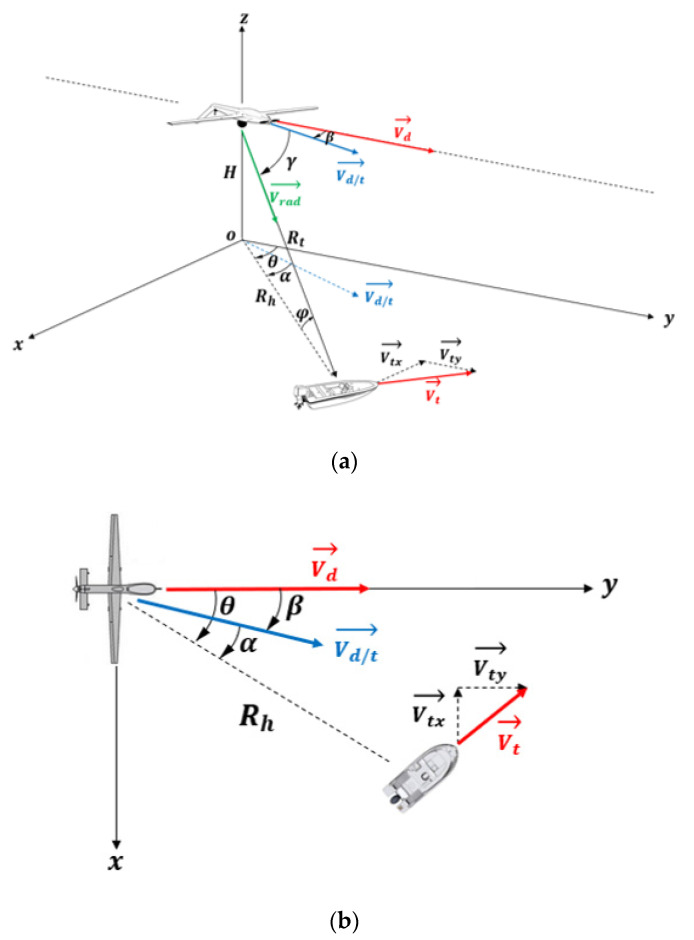
UAV and target motions geometry: (**a**) oblique view, (**b**) top view.

**Figure 14 sensors-22-03263-f014:**
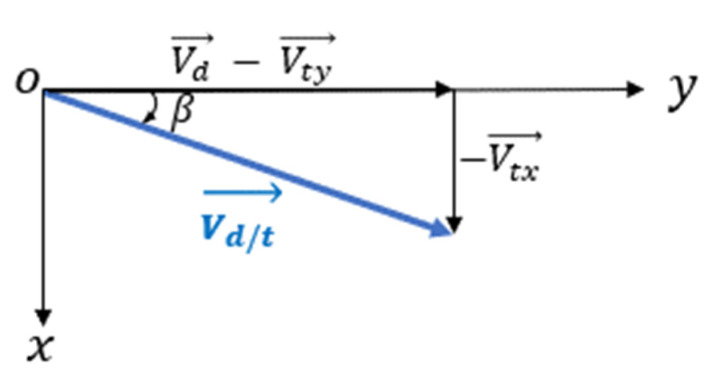
Relative speed of UAV to target.

**Figure 15 sensors-22-03263-f015:**
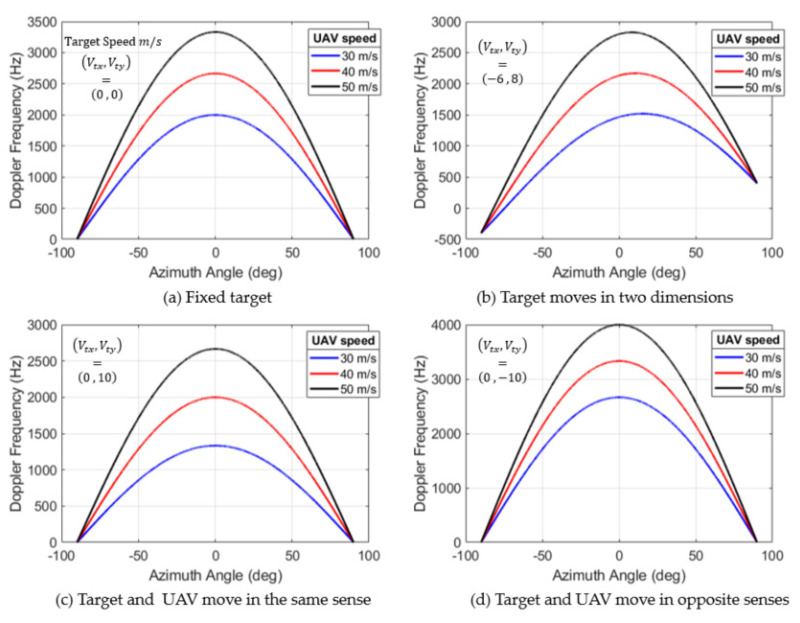
Doppler frequency according to relative movement between the UAV and the target.

**Figure 16 sensors-22-03263-f016:**
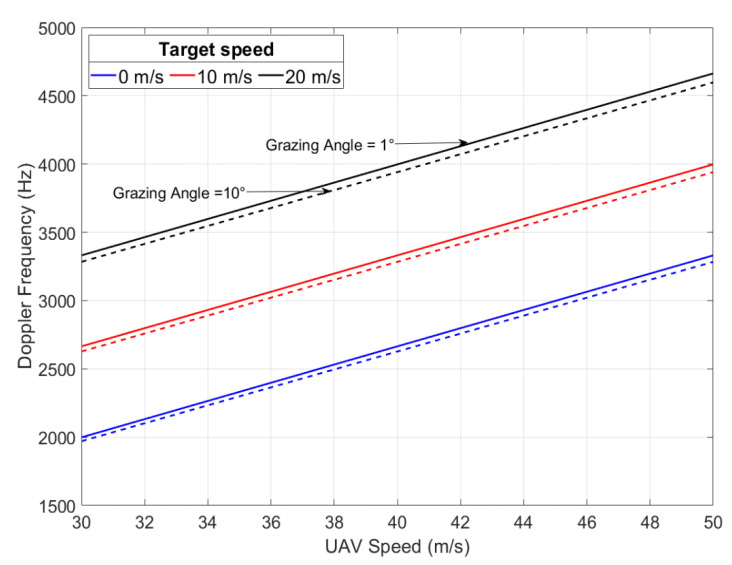
Doppler frequency at θ = 0 versus UAV speed and target speed.

**Figure 17 sensors-22-03263-f017:**
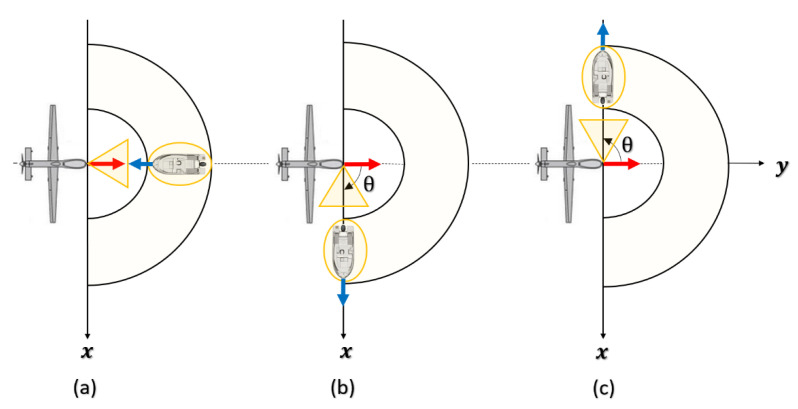
(**a**) Geometry corresponding to the maximum Doppler frequency of target; (**b**,**c**) geometries corresponding to the minimum Doppler frequency of target.

**Table 1 sensors-22-03263-t001:** UAVs Classification according to the US Department of Defense (DoD) [13].

Category	Size	Maximum Gross Takeoff Weight (MGTW) (lbs)	Normal Operating Altitude (ft)	Airspeed (Knots)
Group 1	Small	0–20	<1.200 AGL *	<100
Group 2	Medium	21–55	<3.500	<250
Group 3	Large	<1320	<18.000 MSL **	<250
Group 4	Larger	>1320	<18.000 MSL	Any airspeed
Group 5	Largest	>1320	>18.000	Any airspeed

* AGL = Above Ground Level. ** MSL = Mean Sea Level.

**Table 2 sensors-22-03263-t002:** Attenuation of the radar signal in the X-Ku bands due to gas, cloud, and rain.

Frequency, GHz	Radar Range, km	Attenuation Due to Gas, dB		Attenuation Due to Cloud, dB		Attenuation Due to Rain, dB		
		Water Vapor Density g/m^3^		Liquide Water Density g/m^3^		Rainfall Rate mm/h		
		5	30	0.05	0.5	1	4	16
10	8	0.17	0.93	0.04	0.43	0.16	0.89	4.92
	30	0.65	3.5	0.16	1.6	0.39	2.18	11.40
14	8	0.28	1.97	0.08	0.83	0.48	2.33	10.91
	30	1.07	7.38	0.31	3.13	1.2	5.7	25.28
18	8	0.65	4.62	0.14	1.37	0.92	4.08	17.61
	30	2.43	17.34	0.51	5.15	2.27	9.96	40.80

**Table 3 sensors-22-03263-t003:** Narrower radar beams reduce rain clutter returns.

Azimuth 3 dB beamwidth	0.37°	0.43°	0.33°
Elevation 3 dB beamwidth	11.0°	4.0°	1.8°
Reduction of rain clutter power level	0 dB	3.7 dB	8.4 dB

**Table 4 sensors-22-03263-t004:** The world Meteorological Organization’s Douglas Sea State.

Douglas Sea Scale Degree	Wave Height (Meters)	Characteristics
0	0	Calm (glassy)
1	0 to 0.1	Calm (rippled)
2	0.1 to 0.5	Smooth (wavelets)
3	0.5 to 1.25	Slight
4	1.25 to 2.5	Moderate
5	2.5 to 4	Rough
6	4 to 6	Very rough
7	6 to 9	High
8	9 to 14	Very high
9	Over 14	Phenomenal

## Data Availability

Not applicable.

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
