# Peer review of "Analysis of Small Sea-Surface Targets Detection Performance According to Airborne Radar Parameters in Abnormal Weather Environments"

_sensors, 2022, doi:10.3390/s22093263_

Round 1

Reviewer 1 Report

This manuscript presents a review of previous literature on the modeling of small marine target detection with radars (generally mm-Wave) on UAVs.  I think the manuscript needs more work before it can be considered for publication. 

For one thing, the manuscript does not even mention coherent and incoherent integration of radar scans which can dramatically improve small-target detection in such radar data. 

Several equations (e.g., 12 and 13) are poorly formatted and should be adjusted so that they are not split between more than one line, etc.  

Though equations were given for attentuation in different media, it would also be good to add some discussion on the actual mechanisms for this attentuation, e.g., volumetric scattering.  

The manuscript should also be checked for English errors, which can be found throughout the document.

Author Response

See the answers in the given document and which is titled "sensors-1638297_Responses.to.Editor.Reviewers"

Reviewer 2 Report

This paper reviewed and discussed the problems related to the detection chain of small targets moving on the sea surface via an airborne radar. It is interesting and overall worthy to be published. However, there are some typos and format mistakes for the formula and figures, which should be corrected first. For the section of UAV Radar Measurement Model and Doppler Frequency of Target, only some simulations were presented. Are there any experimental results or related papers? This may be helpful for the readers to understand the presented method.

Author Response

(The authors gave the same response as above.)

Reviewer 3 Report

Background
Along with the rapid development of marine radar, and particularly those carried on aircraft, the detection of small-sized targets which pose an increasing threat has become one of the main areas of interest. However, by considering an observation chain from an aircraft (like a drone) in a maritime environment, with the aim of detecting and tracking of low signal-to-clutter ratio (SCR) targets, one of the important points would be the analysis of the radar system performance according to the radar input parameters, the atmospheric propagation medium, the various sea clutter characterization and the type of targets (RCS, speed…) in this environment. 

-> agree with this background, even though to the best of my knowledge, the authors should be aware of works dealing with UAVs in environment with uncertain current: Adaptive path following for unmanned aerial vehicles in time-varying unknown wind environments; Path planning for unmanned fixed-wing aircraft in uncertain wind conditions using trochoids. These seem relevant due to the uncertain wind in typical sea environments

This review article presents a review of the factors that affect the performance of a UAV's embedded radar and the probability of detection. Among these factors are those related to the radar itself (radar parameters, observation geometry, UAV speed ...), the atmospheric propagation because most of emitted energy is attenuated in this medium and the sea surface which reflects the energy towards the radar direction.
-> I support this work, but if the work is a "review article", the authors should submit it as a review article. Unless I am wrong, this work is currently submitted as a standard reseach article

So, it is necessary to obtain the overall path loss including the anomalous atmospheric environment,
gas attenuation, clouds attenuation, rainfall attenuation, and beam scanning loss. To consider
atmospheric attenuations, ITU-R models are used. On another side, because of spikes and
dynamic variation property, sea clutter is generally described by the statistical distribution with
long tail and by its wider Doppler spectrum. Conventional algorithms such as those based on statistical
model, MTI and MTD processing are often limited, especially for the target of low speed and
low RCS. 
-> I agree with the methods discussed by the authors. But I have aquestion related to the one above. If this is a review article, maybe I am fine like this. But if this is aresearch article, the authors must elaborate on the contribution of this work as compared to the state of the art.

Therefore, sea clutter including empirical and statistical models available are considered to estimate and simulate the impact of radar input parameters, targets RCS and sea state on detection performance. The Doppler frequency of target echo which can be exploited for coherent processing where unwanted return from the sea surface can be suppressed based on the analysis of sea clutter Doppler spectrum characteristics, is described by assuming an adequate scenario of observation geometry.
- Similar comment as before. I would also like to mention that I appreciate the clarity in explaining the various equations. This is also another point for supporting the work

Other comments

The abstract is a bit long

Table I is interesting, but please also provide the appropriate reference to the US Department of Defense in the caption 

If possible, pay attention to having consistent fontize. For example, in eq. (1), one can notice two eqautions with different size

When possible, strive to have the equations on one line, without splitting. For example, eq, (13) maybe can be in one line 

The authors correctly talk about sea-state (SS). is it appropriate to have a table, e.g. as Table I to classify the different sea states?

Please fill the required "Author Contributions:"

46 references suggests that this is not a review article (normally one can have 100+ references in review articles). Therefore, I suggest the authors to remove the "review article" term, and clarify their contribution as compared to the state of the art

Author Response

(The authors gave the same response as above.)

Round 2

Reviewer 1 Report

The authors have adequately addressed my comments from the previous round of review. Therefore, I believe the manuscript is now acceptable for publication after checking for any newly-introduced English/methodological errors.